# Microbial production of megadalton titin yields fibers with advantageous mechanical properties

Christopher H. Bowen[1,6], Cameron J. Sargent [2,6], Ao Wang[3], Yaguang Zhu [1], Xinyuan Chang[1], Jingyao Li[1], Xinyue Mu[1], Jonathan M. Galazka [4], Young-Shin Jun [1], Sinan Keten [3] & Fuzhong Zhang [1,2,5✉]

Manmade high-performance polymers are typically non-biodegradable and derived from petroleum feedstock through energy intensive processes involving toxic solvents and byproducts. While engineered microbes have been used for renewable production of many small molecules, direct microbial synthesis of high-performance polymeric materials remains a major challenge. Here we engineer microbial production of megadalton muscle titin polymers yielding high-performance fibers that not only recapture highly desirable properties of natural titin (i.e., high damping capacity and mechanical recovery) but also exhibit high strength, toughness, and damping energy — outperforming many synthetic and natural polymers. Structural analyses and molecular modeling suggest these properties derive from unique inter-chain crystallization of folded immunoglobulin-like domains that resists inter-chain slippage while permitting intra-chain unfolding. These fibers have potential applications in areas from biomedicine to textiles, and the developed approach, coupled with the structure-function insights, promises to accelerate further innovation in microbial production of high-performance materials.

[1] Department of Energy, Environmental and Chemical Engineering, Washington University in St. Louis, One Brookings Drive, Saint Louis, MO, USA. [2] Division of Biological & Biomedical Sciences, Washington University in St. Louis, One Brookings Drive, Saint Louis, MO, USA. [3] Department of Mechanical Engineering, Northwestern University, Evanston, IL, USA. [4] Space Biosciences Division, NASA Ames Research Center, Moffett Field, CA, USA. [5] Institute of Materials Science & Engineering, Washington University in St. Louis, One Brookings Drive, Saint Louis, MO, USA. [6] These authors contributed equally: Christopher H. Bowen, Cameron J. Sargent. ✉email: fzhang@seas.wustl.edu

Biology is a great source of inspiration for materials design, as nature is capable of producing many high-performance, biodegradable materials from renewable feedstock through low-energy, aqueous processes[1,2]. Examples include the exceptionally tough insect silks[3], underwater adhesive mussel byssus[4], compression-resistant abalone nacre[5], and highly elastic insect resilin[6]. In many cases, these natural materials can outperform the best available petroleum-based alternatives[1,3,4]. Unfortunately, many of these high-performance natural materials cannot be easily harvested from their native sources, and their natural biosynthetic processes are often impossible to harness for scalable production as they are produced in limited quantity by slow-growing organisms[2,3]. Thus, engineered microbial production strategies are needed to facilitate the practical use and development of these high-performance, renewable materials.

While engineered microbes have been used successfully for scalable production of a great range of small-molecule compounds[7–11], the direct microbial production of polymeric materials with high mechanical performance has remained very limited[1]. Many high-performance natural materials are protein-based and derive their superior mechanical performance from hierarchical assemblies of ultra-high molecular weight (UHMW) proteins with highly repetitive amino acid sequences[1–3,12,13]. These UHMW, repetitive proteins are exceedingly difficult to produce in microbes due to genetic instability, low translation efficiency, and metabolic burden[14,15]. The muscle protein titin, for example, endows muscle tissue with a combination of passive strength, damping capacity, and rapid mechanical recovery derived from titin's UHMW (>3 MDa) and highly repetitive sequence comprising hundreds of folded immunoglobulin (Ig) domains (Fig. 1a)[16–18]. While these appealing mechanical properties have inspired many efforts to engineer titin-like materials[19–23], titin's massive size and repetitive sequence have largely restricted these efforts to the production of titin-mimetic organic polymers rather than more environmentally-friendly protein-based materials (PBMs)[20]. In fact, previous work involving recombinant production of titin proteins has only succeeded in expressing relatively short fragments (usually 8–12 Ig domains)[24–27], and to our knowledge, no prior efforts have been made to produce a macroscale material from microbially produced titin proteins.

In this work, we address these challenges to potentiate microbial production of UHMW protein polymers for the innovation of renewable high-performance materials. We employ a synthetic biology approach to mitigate the challenges of genetic instability and low translational efficiency through in vivo protein polymerization catalyzed by split-inteins (SI) in *Escherichia coli* (Fig. 1b). In this manner, we microbially produce titin polymers with megadalton MW and subsequently develop an aqueous process to spin the resulting polymers into high-performance titin-based fibers that exhibit an intriguing combination of desirable mechanical properties.

## Results

**Engineered in vivo polymerization yields an UHMW titin polymer**. To enable efficient microbial production of UHMW titin proteins, we genetically fused the C- and N-terminal halves of a fast-reacting SI pair (gp41-1)[28] to the N- and C-termini, respectively, of a relatively short titin subunit containing four Ig domains, yielding a chimeric protein Int$^C$-4Ig-Int$^N$ (Fig. 1b). SIs catalyze spontaneous splicing reactions, covalently linking their fusion partners via a peptide bond and leaving only a few residues (≤ 6) at the ligation site with minimal effect on the properties of the resulting product[28–32]. We hypothesized that expressing this chimeric protein in *E. coli* would, through multiple rounds of

intracellular SI-catalyzed ligation of 4Ig subunits, produce UHMW titin polymers (Fig. 1b). To minimize intramolecular ligation and cyclization, we chose to polymerize a relatively rigid subunit of four Ig domains (I67–I70, hereafter 4 Ig) from the I-band of the rabbit soleus muscle titin (Fig. 1b). A previously reported crystal structure of 4 Ig suggests structural rigidity and shows the N- and C-termini of the subunit spatially opposed and separated by approximately 16.4 nm (Supplementary Fig. 1)[18].

The chimeric protein Int$^C$-4Ig-Int$^N$ and the 4 Ig without SIs were expressed separately in *E. coli* shake flask cultures. SDS-PAGE analysis showed that the cells expressing the 4Ig monomer produced a single band at the expected molecular weight of 43 kDa, while the Int$^C$-4Ig-Int$^N$ expressing cells yielded a cluster of UHMW products up to and above 460 kDa. (Supplementary Fig. 2a). Analytical size exclusion chromatography (SEC) confirmed that the purified titin polymers are indeed UHMW, with approximately 20% of the eluted species exceeding the column fractionation limit of 5 MDa (Supplementary Fig. 2b). Even without considering this UHMW fraction, the mass average MW ($M_w$) of the titin polymer was estimated to be 2.4 MDa. Meanwhile, SEC analysis of the purified monomer under identical conditions revealed a sharp eluent peak that corresponded roughly to the expected MW for the monomer (43 kDa).

Circular dichroism (CD) was used to examine the secondary structure of the purified and refolded polymer, yielding spectra that are qualitatively similar to those previously reported for natural titin protein extracts (Fig. 2a)[33]. Further deconvolution and fold recognition by the BeStSel deconvolution tool[34] suggest a high degree of anti-parallel β-sheet structure (Fig. 2a, inset) and an immunoglobulin-like topology based on six of the top ten closest structures in the eight-dimensional BeStSel secondary structural space (Supplementary Note 1). Additionally, scanning transmission electron microscopy (STEM) of the purified titin polymer (Fig. 2b, Supplementary Fig. 3) showed the presence of numerous nanoscale fibrils with apparent chain-of-beads structures and cross-sectional diameters (6.1 ± 1.2 nm, Supplementary Fig. 4) that were similar to those previously observed for natural titin proteins[18]. Together, these results suggest that our microbial production system can synthesize UHMW titin polymers with a substantial degree of folded structures similar to natural Ig domains (hereafter called Ig-like structures).

**Wet-spinning UHMW titin yields monofilament fibers**. To explore practical uses of our microbially produced UHMW titin polymer, we next sought to process the polymer into macroscale monofilament fibers. It is known that networks of inter-chain crystallites, serving as non-covalent cross-links embedded in an amorphous matrix, can provide high strength and toughness at the macroscale[35]. We hypothesized that if the polymer were refolded from a denatured state at high concentration, the robust folding of Ig domains might favor the formation of a network of inter-chain β-sheet crystals. Thus, we dissolved the titin polymer at a high concentration in the denaturing solvent hexafluoroisopropanol (HFIP) and then extruded the resulting dope through a narrow-bore needle into water with the aim of driving rapid titin refolding during fiber formation. Indeed, fibers were formed, which we subsequently subjected to post-spin draw to promote axial alignment of the expected crystalline domains[36]. Both light microscopy and scanning electron microscopy (SEM) revealed cylindrical, monofilament fibers with highly consistent diameters of approximately 10 μm (Fig. 2c, d, Supplementary Fig. 5, Supplementary Table 1). Based on the observed spinning efficiency and current protein yields, we estimate that 1 L of shake flask batch culture could yield ~250 m of fiber in a continuous spinning process.

Fourier-transform infrared spectroscopy (FT-IR) analysis of the UHMW titin fibers confirmed a substantial percentage of β-sheet

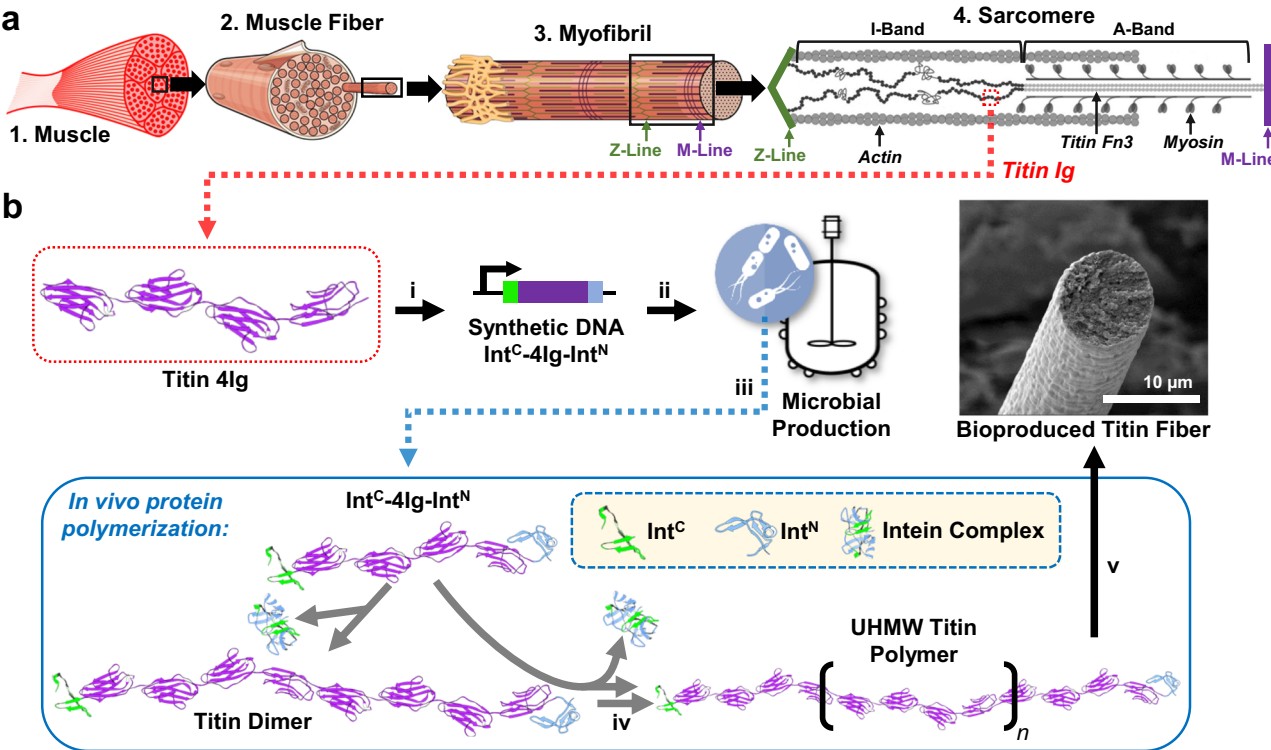

**Fig. 1 The multi-scale structure of muscle and schematic representation of SI-based polymerization of the titin protein in *E. coli*. a** Muscle tissue (1) is composed of specialized, elongated (>1 cm) cells called muscle fibers (2). Muscle fibers are packed with proteinaceous myofibrils (3) that span the entire length of the cell. Myofibrils are composed of repeating stacks of chemically controllable, contractile elements called sarcomeres. (4) Sarcomeres are composed primarily of three proteins: actin, myosin, and titin. Titin spans half the length of the sarcomere, anchoring the opposing Z- and M-lines, and consists of hundreds of repeating immunoglobulin (Ig) domains that are integral to the passive strength (i.e., resistance to deformation without energy input), damping capacity, and mechanical recovery of the macroscopic muscle fiber. The images of muscle fiber and myofibril are modified from the OpenStax Anatomy and Physiology Textbook Version 8.25, Published May 18, 2016 (OpenStax, CC BY 4.0 https://creativecommons.org/licenses/by/4.0) and the image of the sarcomere is modified from Giganti, D., Yan, K., Badilla, C.L. et al. Disulfide isomerization reactions in titin immunoglobulin domains enable a mode of protein elasticity. Nat Commun 9, 185 (2018). 10.1038/s41467-017-02528-7, (David Giganti, Kevin Yan, Carmen L. Badilla, Julio M. Fernandez & Jorge Alegre-Cebollada, CC BY 4.0 https://creativecommons.org/licenses/by/4.0). **b** To facilitate the production of UHMW titin polymer in vivo, a relatively small, genetically stable, 41.3 kDa rabbit soleus titin protein-coding sequence (4 Ig; purple) was flanked by complimentary SIs, gp41-1$^C$ (Int$^C$; green) and gp41-1$^N$ (Int$^N$; blue) (i). DNA sequence-recoded Int$^C$-4Ig-Int$^N$ was produced in an engineered *E. coli* host under the control of inducible promoter P$_{LacO-1}$ (ii). The SI-flanked monomer protein was overexpressed in bioreactor cultures (iii) and polymerized intracellularly through successive rounds of SI-catalyzed intermolecular ligation to produce UHMW titin (iv). Purification and processing yielded microbially produced titin fibers that recapture the damping capacity and mechanical recovery of muscle along with high strength and toughness (v). Titin 4 Ig and SI structures were acquired using PBD accession numbers 3B43 and 6QAZ, respectively.

secondary structure in the fiber (Fig. 2f, Supplementary Fig. 6). Deconvolution of the amide I peak allowed us to estimate a β-sheet content of ~28%, with no significant difference between fibers with or without a post-spin draw (Fig. 2e, f). We then used polarized Raman spectromicroscopy to examine the alignment of β-sheets along the fiber axis. The as-spun fibers showed no orientation sensitivity in the amide I and II bands (Supplementary Fig. 7a), suggesting that the β-sheets formed during fiber spinning are randomly oriented. Conversely, polarized Raman spectra of the post-spin drawn fibers revealed a high degree of orientation sensitivity in the amide I and II bands, which is characteristic of axially aligned β-sheet structures (Fig. 2g, Supplementary Fig. 7b)[37]. These results confirm that the spinning process produces a fiber that is rich in β-sheets with initially random orientation, while the subsequent post-spin draw helps to align these β-sheets along the fiber axis.

**X-ray diffraction suggests Ig-like molecular structure.** To further examine the structure of the post-spin drawn UHMW titin fibers, we employed synchrotron-based wide-angle X-ray diffraction (WAXD). Two-dimensional diffraction images revealed two broad but distinct equatorial reflections perpendicular to the fiber axis,

along with substantial amorphous components (Fig. 2h–k, Supplementary Fig. 8a), characteristic of a semi-crystalline material[36,38]. The deconvolution of peaks suggests approximately 18% crystallinity (see Methods, Supplementary Tables 2, 3). To study the crystalline orientational order, we analyzed the azimuthal 1D profiles of the two primary reflections[39–41] and determined an orientation parameter $f_{crystal} = 0.76$ for the crystalline portion of our titin fibers (see Methods, Fig. 2j, k, Supplementary Tables 2, 3). This indicates a substantial axial crystallite alignment, in agreement with the polarized Raman spectromicroscopy analysis (Supplementary Fig. 7). The average crystallite size was estimated through further deconvolution of peaks and application of the Scherrer equation[42]. In this manner, both the average crystallite size along the a-axis (inter-sheet axis) and the d-spacing were calculated to be 1.08 nm (see Methods, Fig. 2h, i, Supplementary Fig. 8b, Supplementary Tables 2, 3). This is in close agreement with the average distance (approximately 1.01 nm) between opposing β-sheets of individual Ig domains in the previously reported I67–I70 crystal structure (Supplementary Fig. 1)[18]. The fact that both the d-spacing and average crystallite size along the a-axis are 1.08 nm suggests that the titin fiber likely contains β-crystals of two β-sheets, similar to the structure of the native titin Ig domain.

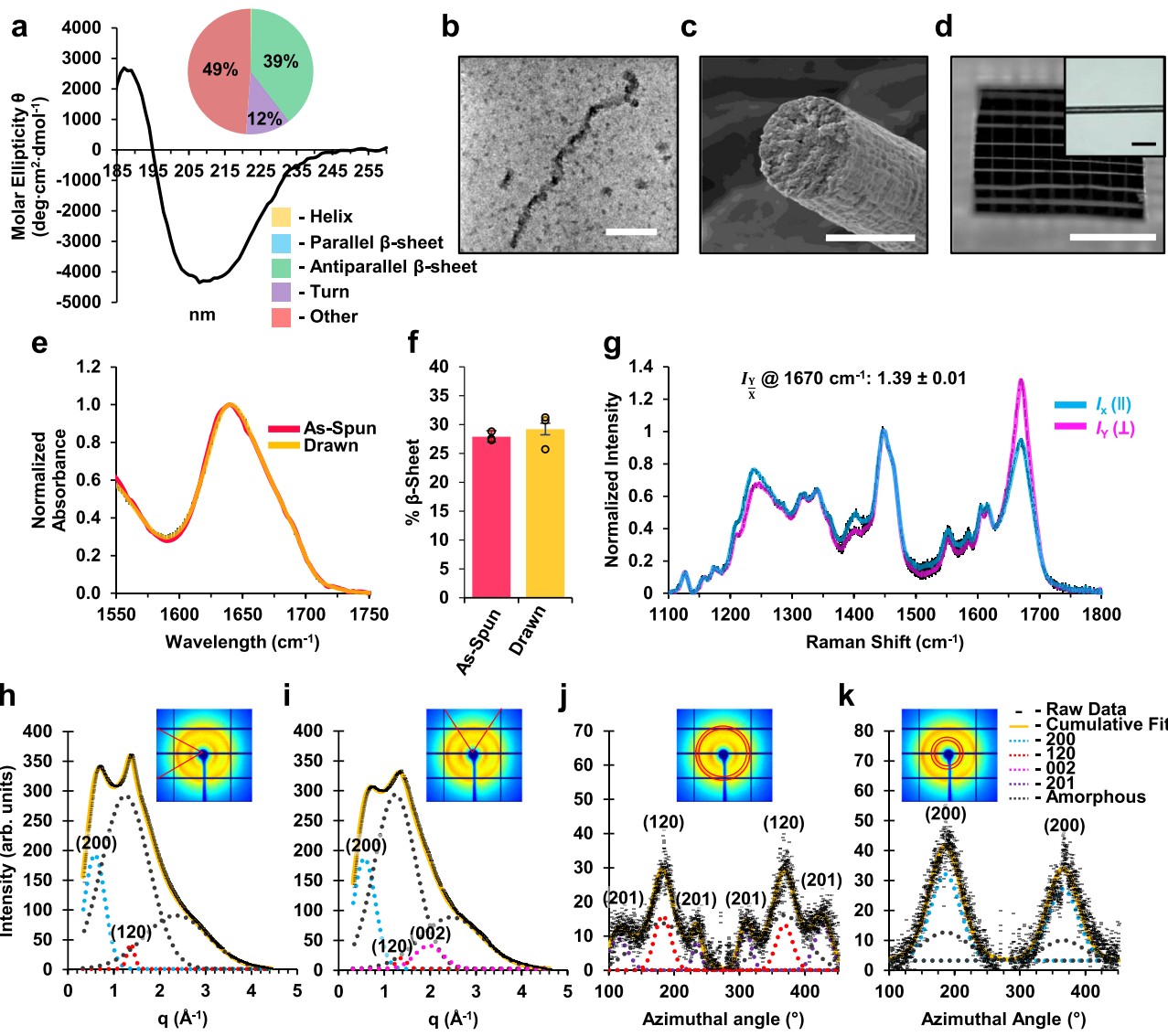

**Fig. 2 Structural analyses of microbially produced UHMW titin protein and processed monofilament fibers. a** Circular dichroism spectrum for purified titin polymer in water. The inlaid pie graph indicates the results of spectral deconvolution by the BeStSel program[34]. **b** STEM image of purified titin polymer. The scale bar is 50 nm. This image is representative of dozens of similar polymer molecules observed using STEM, 42 of which were selected for diameter measurements. **c** SEM image of a fracture cross-section of the spun titin fiber. The scale bar is 10 μm. This image is representative of 12 fibers that were observed using SEM. **d** Image of a textile net woven from the spun titin fibers. The white scale bar is 0.5 cm. Inset is a light microscopy image of an individual titin fiber, representative of all of the images taken for diameter measurements. The black scale bar is 40 μm. **e** FTIR analysis of as-spun and post-spin drawn UHMW titin polymer fibers. Averages of normalized spectra for each condition were overlaid. **f** Deconvolved β-sheet content of titin polymer fibers. For each fiber state, percentages were averaged for FTIR spectra acquired from three separate fibers. Error bars are the standard deviation of the three peak area calculations (see Methods, Supplementary Fig. 6). **g** Raman spectra of post-spin drawn titin polymer fibers oriented perpendicular ($I_Y$; pink line) or parallel ($I_X$; blue line) to the polarization of the incident laser. Spectra shown are the average of spectra acquired from three separate fibers. Standard deviations of the three measurements at each Raman shift are shown as black bars. The average ratio of the amide I peak (1670 cm$^{-1}$) intensity at 0° to that at 90° is shown above the spectrum as a measure of orientation sensitivity (see Methods, Supplementary Fig. 7). **h**–**k** Synchrotron-based wide-angle X-ray diffraction analysis of spun titin polymer fibers. Insets show the area selected for radial (**h**, **i**) or azimuthal (**j**, **k**) integration. **h** 1D radial intensity profile along the equator, with Gaussian fits for the (120) equatorial peak (dotted red), (200) equatorial peak (dotted blue), and two amorphous components (dotted gray). **i** 1D radial intensity profile along the meridian, with Gaussian fits for the (120) meridian peak (dotted red), (200) meridian peak (dotted blue), (002) peak (dotted pink), and two amorphous components (dotted gray). **j**, **k** Intensity as a function of the azimuthal angle at the radial position of the equatorial (120) peak (**j**) and (200) peak (**k**). The peaks are fitted as sums of two Gaussians, corresponding to crystalline (narrow) and amorphous (broad) distributions. In figure (**j**), small subsidiary peaks due to residual intensity from the (201) reflections (dotted purple) were treated as individual Gaussian functions. Source data are provided as a Source Data file.

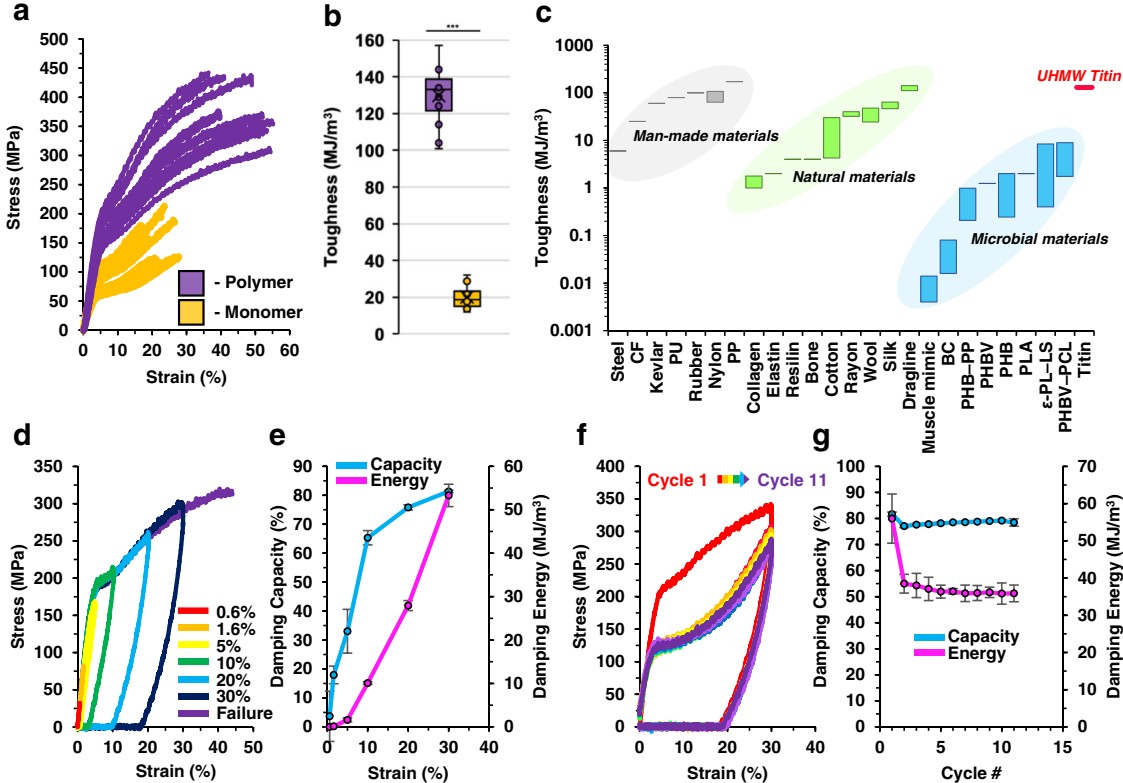

**Fig. 3 Mechanical testing of fibers spun from microbially produced UHMW titin reveals high toughness, damping capacity, and mechanical recovery reminiscent of natural muscle fibers. a** Stress-strain curves from tensile tests of 14 microbially produced UHMW titin fibers (polymer; purple) and the low MW, 4Ig titin (monomer; gold). **b** Box-plots displaying toughness measures extracted from the stress-strain curves for polymer (purple) and monomer (gold) fibers (n = 14 fibers for each protein; horizontal lines denote, from top to bottom, upper fence, Q3, median, Q1, and lower fence; × denotes mean; other data indicated with circles). ***$P = 1.6 \times 10^{-19}$, unpaired two-tailed t-test. **c** Toughness of microbially produced (blue), natural (green), and man-made (gray) materials compared to that of the UHMW titin fibers produced in this work (red). **d** Loading/unloading curves for microbially produced UHMW titin fibers acquired at increasing strains from 0.6–30%. **e** Average calculated damping capacity (blue curve) and damping energy (pink curve) at each strain tested in (**d**). Error bars are the standard deviation of the three fiber samples tested at each strain. **f** Stress-strain curves for microbially produced UHMW titin fibers subjected to 11 consecutive loading/unloading cycles with one minute of humid (95% RH) air treatment between cycles. The stress-strain curve of the first round is colored red. Following cycles use other colors. **g** Average calculated damping capacity (blue curve) and damping energy (pink curve) over consecutive cycles with humid air treatment between cycles. Error bars are the standard deviation of the values measured at each cycle number for the three fiber samples that were tested. Source data are provided as a Source Data file.

Meanwhile, the average crystallite size along the b-axis (inter-strand axis) was calculated to be 2.91 nm, with an inter-strand d-spacing of 0.46 nm (Supplementary Fig. 8b, Supplementary Tables 2, 3). The observed inter-strand d-spacing in our titin fibers agrees with the average inter-strand distances of anti-parallel β-strands in the crystal structure of I67–I70 (~0.46 nm, Supplementary Fig. 1)[18]. However, the calculated b-axis width of 2.91 nm suggests an average of 6 β-strands per β-sheet, more than what is observed in the native titin Ig domains (3–4 β-strands per β-sheet, average width 1.46 nm; Supplementary Fig. 1), suggesting that some of the crystals may form from side-by-side packing of pairs of Ig-like domains.

**UHMW titin fibers exhibit excellent mechanical properties.** We next investigated the mechanical properties of our microbially produced UHMW titin fibers to determine whether they could reproduce the desirable mechanics of natural muscle fibers. Tensile testing revealed high strength (378 ± 41 MPa), modulus (4.2 ± 0.6 GPa), extensibility (47 ± 7%), and toughness (130 ± 15 MJ/m$^3$) (Fig. 3a, b, Supplementary Fig. 9). Both the strength and toughness of these fibers far exceed values measured for muscle fibers and individual myofibrils[16,43,44]. Furthermore, these toughness measures even exceed those of many of the toughest synthetic and natural materials, and far exceed those of traditional microbial materials (Fig. 3c)[23,45–48]. SEM images of fibers after fracture indicate a

uniform, densely packed microscale morphology (Supplementary Fig. 5). Meanwhile, fibers spun in an identical manner from the low MW 4Ig exhibited dramatically lower strength (−60%), modulus (−38%), breaking strain (−57%), and toughness (−85%) when compared to the UHMW fibers (Fig. 3a, b, Supplementary Fig. 9). This result confirms that the high mechanical performance of our fibers partially results from the previously unobtainable UHMW of the microbially produced titin polymer, highlighting the value of the protein polymerization strategy.

While it is generally accepted that the MW of a polymer can greatly affect the mechanical properties of the resulting material[13,36,49], we sought to provide further evidence for a correlation in titin fibers and to confirm that the MW was a primary factor in the observed performance of the UHMW titin fibers. To produce materials of a precise MW, we constructed genes that contained two and three repeats of the 4 Ig sequence, referred to as 8 Ig and 12 Ig, respectively. These proteins were then expressed, purified, and spun into fibers. Tensile testing of these fibers revealed a strong positive correlation between MW and mechanical properties (Supplementary Fig. 9). Still, the UHMW fibers offered far greater strength and toughness (79% and 85%, respectively) than the 12 Ig fibers, while eliminating risks of genetic instability that could otherwise diminish bioproduction yields, further demonstrating the value of the in vivo polymerization platform.

Within muscle fibers, the natural titin protein behaves as a resilient material at low strain, capable of reversible deformation without energy loss, and as an energy damping material at higher strain, capable of dissipating energy to prevent myofibril damage due to over-extension[16,18,20,50]. This combination of and transition between resilient and energy damping states makes titin uniquely suited for the crucial biological functions of skeletal and cardiac muscle. To investigate whether the microbially produced UHMW titin fibers can recapitulate this desirable combination of properties, we performed cyclic force loading experiments over a range of increasing strains. We found that at up to 1.6% strain, the fibers can be stretched elastically with relatively high resilience and low damping capacity (17.9 ± 3.0%) and damping energy (0.1 ± 0.0 MJ/m$^3$) (Fig. 3d, e). However, as we increased fiber extension beyond 1.6%, damping capacity increased rapidly, reaching up to 81.3 ± 0.4% at 30% strain (Fig. 3d, e). As expected, damping energy also rapidly increased with elongation, reaching a maximum of 53.3 ± 2.6 MJ/m$^3$ at 30% strain. This damping capacity exceeds previous measures of the damping capacity of both myofibrils and natural single-molecule titin, which have been measured at around 60%[43,44], as well as many high-damping synthetic and natural materials (Supplementary Fig. 10)[23]. When tested at increasing strains, fibers made from the 4 Ig monomer also demonstrated resilience at low strains followed by a rapid increase in damping capacity as the strain increased; however, the damping energy measured for the monomer fibers was greatly reduced relative to those of the polymer fibers likely due to the decreased MW of the constituent proteins (Supplementary Fig. 11a, b).

During relaxation of natural single-molecule titin after high extension, unfolded Ig domains are believed to refold, enabling a form of self-repair and recovery of mechanical properties[16–18]. To examine whether our microbially produced UHMW titin fiber has similar properties, we subjected the fibers to repeated rounds of loading-unloading cycles approaching the fiber-breaking strain. After drawing fibers to 30% strain at ambient humidity (45% RH) and relaxing back to 0% strain, we observed an apparent permanent set at approximately 20% strain (i.e., fibers could only rapidly recover about 33% of the total deformation) (Fig. 3f, Cycle 1). Consequently, to better mimic the aqueous environment of natural titin in the muscle, we briefly exposed the stretched fibers to high humidity air (95% RH) following relaxation, whereupon we observed that the fibers rapidly contracted back to their original lengths (0% strain). Under these conditions, we found that the second round of loading-unloading exhibited only a slight, 5% decrease in damping capacity and a 15 MJ/m$^3$ decrease in damping energy compared to the first cycle (Fig. 3f, g). Additional loading-unloading cycles resulted in no further reduction of damping capacity and only an additional 3 MJ/m$^3$ decrease in damping energy over a successive 10 cycles (Fig. 3f, g). Fibers made from the titin monomer also demonstrated high damping capacity (~60%) and humidity-driven recovery when pulled to a near maximal strain (in this case 12%) over several cycles of loading, unloading, and humidity treatment (Supplementary Fig. 11c, d). Despite the damping energy of the monomer fibers being much lower (~4.7 MJ/m$^3$) due to differences in MW, the relatively high damping capacity suggests that the humidity-driven recovery of the fibers may be mediated by the refolding of Ig domains. Thus, with 95% RH treatment, the microbially produced titin fibers can rapidly recover mechanical properties in a manner reminiscent of natural titin and muscle fiber. This intriguing regenerative behavior, along with the material's combination of high damping capacity, strength, and toughness, suggest a broad range of potential applications for these fibers in areas such as anti-ballistic materials, netting, sutures, and tissue engineering[51–54].

**MD simulation suggests a unique energy dissipation mechanism.** To help elucidate the possible energy dissipation mechanisms that result in the observed mechanical properties of the UHMW titin fibers, we carried out molecular dynamics (MD) simulations to study the behavior of modeled titin fibers under tensile deformation. While the stretching behavior of single titin chains has been widely studied, both with AFM experiments[17,55–59] and steered MD (SMD) simulations[60,61], the macroscale biosynthetic titin fiber described in this work required a unique model featuring a network of tightly packed titin chains and periodic boundary conditions to provide a better representation of the bulk properties.

Our initial fiber molecular model was built according to the results of our structural analyses, which indicated that pairs of Ig-like domains from adjacent titin polymer chains are packed side-by-side and axially aligned (see Methods, Supplementary Fig. 12, Supplementary Note 3). A uniaxial tensile strain was then applied to the simulated fiber, and a tensile stress-strain curve was measured. The resulting modeled curves agreed well with the experimental results (Fig. 4a), showing an initial elastic deformation up to approximately 5% strain, with a modulus of 3.6 ± 0.2 GPa (compared to 4.2 ± 0.6 GPa for the experimental titin fibers), followed by a period of plastic deformation and strain hardening with peak stress of 378 ± 17 MPa (compared to 378 ± 41 MPa for the experimental titin fibers). While the real fibers exhibited a sudden failure at an average strain of 47%, the model showed a relatively gradual decrease in stress beyond 45%. This difference may be attributed to the size effects of the small volume of the MD simulation relative to that of the experimental specimen. Specifically, in the macroscale fibers, the presence of defects and strain localization upon yielding are expected to result in a more sudden failure than that observed at the molecular scale[62].

Observation of structural and energy changes throughout the course of the simulation suggests that there is little change in the structure or relative position of Ig domain pairs up to 10% strain (Fig. 4b), with tensile stress distributed evenly across the structure (Fig. 4c, d). While there is a slight disruption of intra-fibril Van der Waals interactions in this regime (Supplementary Fig. 13b), there is no substantial change in electrostatic interactions or hydrogen bonding (Supplementary Fig. 13c–f). In the subsequent plastic deformation and strain-hardening regime (10–50% strain), the linker regions between Ig domains become fully extended (Fig. 4b) and stress is accumulated within Ig-like domains (Fig. 4c, d), disrupting intra-fibril hydrogen bonds (primarily backbone-backbone, Supplementary Fig. 13d–f) and electrostatic interactions (Supplementary Fig. 13c) in addition to the continued disruption of intra-fibril Van der Waals interactions (Supplementary Fig. 13b). Interestingly, throughout this regime, overall inter-fibril interactions actually increase (Fig. 4e), driven by increases in inter-fibril electrostatic interactions (Supplementary Fig. 13c) and inter-fibril hydrogen bonding (Supplementary Fig. 13d–h). This suggests that fiber pulling actually drives the annealing of the inter-fibril pairs of Ig-like domains, further strengthening the fibers. Finally, at high strains (50–80%), some Ig-like domains undergo substantial unfolding (Fig. 4b, c), with a major disruption of intra-fibril hydrogen bonding within these sacrificial domains (Fig. 4f, red line). Remarkably, this unfolding occurs before any apparent slippage between adjacent titin polymer chains (Fig. 4b), with a continued increase in overall inter-fibril interactions (Fig. 4e) that includes inter-fibril electrostatic interactions and hydrogen bonding (Supplementary Fig. 13c–f). Furthermore, due to the unfolding of the sacrificial Ig domains, stress is reduced in other Ig domains, allowing them to relax and regain some stabilizing intramolecular hydrogen bonds (Fig. 4d, f, yellow line). While unfolding is only observed in the model above 50% strain, it is likely that such Ig domain unfolding would occur within the macroscale fiber at lower strains in areas of concentrated stress, thus contributing to the overall energy damping and toughness of the macroscale titin fibers.

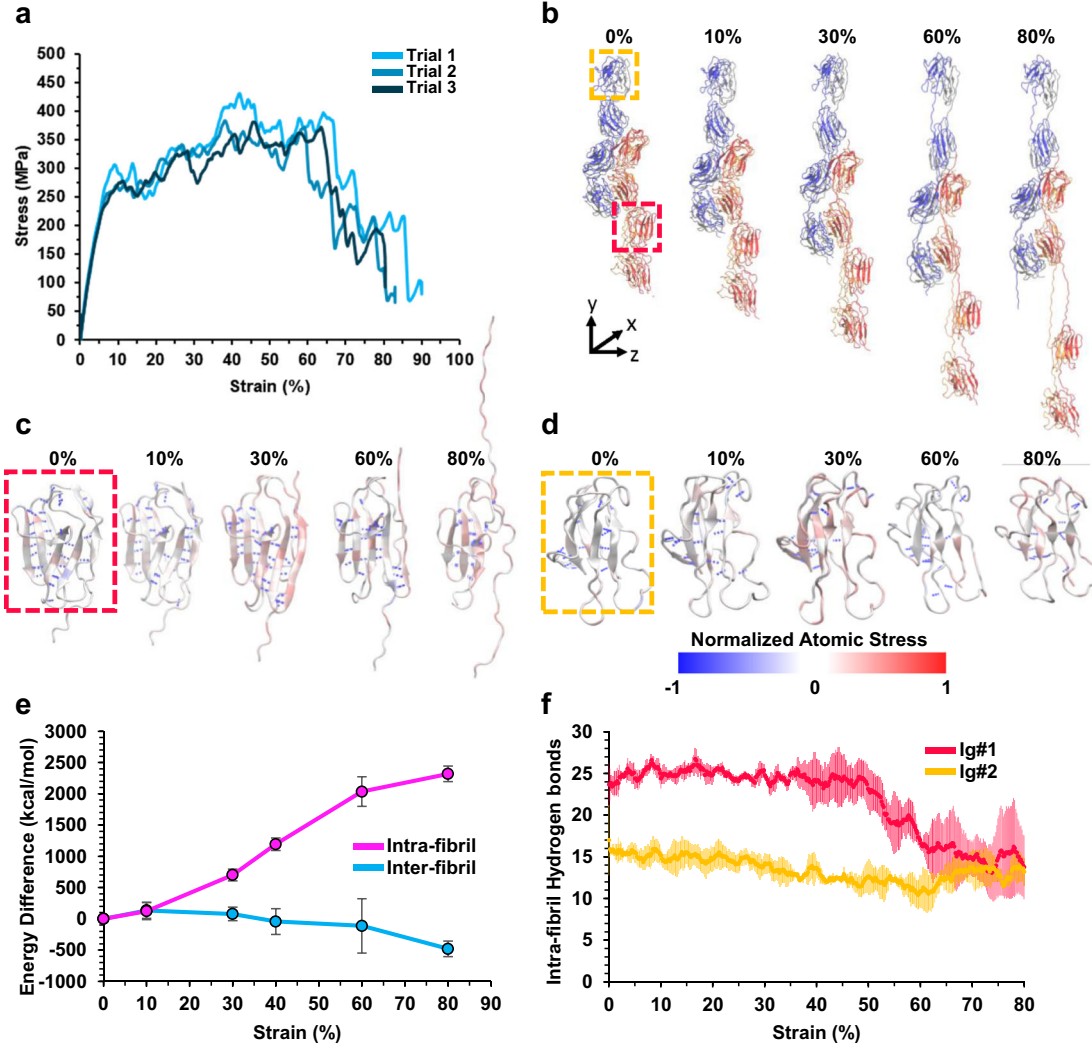

**Fig. 4 Molecular dynamics simulation of uniaxial tensile testing of a model titin fiber. a** Representative uniaxial tensile stress-strain curves of the model titin fiber. **b** Snapshots of the molecular dynamics simulation of the titin fiber under tensile deformation. **c, d** Normalized atomic stress along the y-axis during extension of the titin fiber from 0 to 80% strain. Selected Ig-like domains are shown in dashed boxes. The Ig-like domain in the red and yellow boxes in (**b**) are shown in (**c, d**), respectively. **e** Average changes in intra- and inter-fibril (pink and blue, respectively) non-bonded energies (including Van der Waals, electrostatic, and hydrogen bonds) over the course of the simulation. Error bars are the standard deviation of three trials. **f** Average total number of intra-fibril backbone-backbone hydrogen bonds in the two selected Ig-like domains in (**b–d**) over the course of the tensile test. Error bars (shown as red or yellow bars) are the standard deviation of three trials. Source data are provided as a Source Data file.

Together, these modeling results suggest that the excellent mechanical properties of the microbially produced UHMW titin fibers may originate from a unique inter-fibril pairing of folded Ig-like domains. Such inter-chain, non-covalent crosslinking through folded, stretchable domains has rarely been explored in either organic polymeric materials or other microbially produced fibers.

## Discussion

By harnessing the biosynthetic power of microbes, this work has produced a novel high-performance material that recaptures not only the most desirable mechanical properties of natural muscle fibers (i.e., high damping capacity and rapid mechanical recovery) but also high strength and toughness, higher even than that of many manmade[19–23,46,48] and natural high-performance fibers[3,45–47]. To our knowledge, this is the first example of an engineered macroscale material produced from titin. The fiber's highly desirable

combination of mechanical properties, sustainable production process, and biodegradability make it an excellent candidate for environmentally friendly applications in a range of fields from biomedicine to commercial textiles (e.g., anti-ballistic materials, netting, sutures, and tissue engineering)[51–54]. Our structural analyses suggest that these UHMW titin fibers contain axially aligned, side-by-side pairs of Ig-like domains. Molecular dynamics simulations indicate that such a network of non-covalent crosslinking through Ig-like domains can strongly resist chain slippage, while permitting domain unfolding, giving rise to the observed combination of mechanical properties. Materials production through such non-covalent crosslinking of folded, stretchable proteins has rarely been explored, and these results could potentially inform the design of other high-performance materials that exploit this paradigm to yield a range of intriguing macroscale material properties. Thus, this work represents a significant expansion of the range of products accessible through engineered microbial synthesis, moving from primarily small molecules, peptides, therapeutic proteins, and industrial enzymes toward

effective direct production of high-performance materials. It is likely that the biosynthetic strategy developed here can be applied to other proteins with robust folding properties, yielding novel, high-performance materials with an expanded range of properties and offering an increasing variety of sustainable alternatives to traditional petroleum-based polymers.

## Methods

**Strains and growth conditions**. *E. coli* NEB 10-beta (NEB10β) was used for all plasmid cloning and protein production. For all cloning, *E. coli* strains were cultured in Terrific Broth (TB) containing 24 g/L yeast extract, 20 g/L tryptone, 0.4% v/v glycerol, 17 mM $KH_2PO_4$, and 72 mM $K_2HPO_4$ at 37 °C with appropriate antibiotics (50 μg/mL kanamycin). M9 glucose medium with tryptone supplement (2% w/v glucose, 1× M9 Salts, 75 mM MOPS pH 7.4, 12 g/L tryptone, 5 mM sodium citrate, 2 mM $MgSO_4$ $7H_2O$, 100 μM $FeSO_4$ $7H_2O$, 100 μM $CaCl_2$ $2H_2O$, 3 μM thiamine, 1× micronutrients [40 μM $ZnSO_4$ $7H_2O$, 20 μM $CuSO_4$ $5H_2O$, 10 μM $MnCl_2$ $4H_2O$, 4 μM $H_3BO_3$, 0.4 μM $(NH_4)_6Mo_7O_{24}$ $4H_2O$, and 0.3 μM $CoCl_2$ $6H_2O$]) was used for protein production in bioreactors.

**Chemicals and reagents**. Unless otherwise noted, all chemicals and reagents were obtained from MilliporeSigma. Plasmid purification and gel extraction kits were purchased from iNtRON Biotechnology. FastDigest restriction enzymes and T4 DNA ligase were purchased from Thermo Fisher Scientific and used for all digestions and ligations following manufacturer protocols.

**Construction and expression optimization of titin monomer and polymerization cassettes**. The amino acid sequence of rabbit soleus titin domains I67-70 was obtained from a recent publication[18], and the coding sequence was computationally optimized for *E. coli* expression using DNA 2.0 (ATUM) (Supplementary Table 4)[30]. The resulting optimized sequence was synthesized as a gBlock fragment by Integrated DNA Technologies. The sequence was then inserted between the KpnI and Kpn2I restriction sites of modified BglBricks[63] vectors containing gp41-1C and gp41-1N SIs under the control of a $P_{BAD}$ promoter or a $P_{LacO1}$ promoter, yielding plasmids p-1-4XT-1$_B$ and p-1-4XT-1$_L$, respectively (Supplementary Table 5). Additionally, the optimized titin sequence was inserted between the KpnI and Kpn2I restriction sites of a modified BglBricks vector containing no SIs under control of a $P_{LacO1}$ promoter, yielding plasmid p-4XT (Supplementary Table 5). To construct the 8Ig titin plasmid, PCR was first used to amplify the optimized 4XT sequence from p-4XT, adding a Kpn2I restriction site to the 5′ end and maintaining a stop codon and a BamHI site at the 3′ end of the amplicon (Supplementary Table 6). This amplicon was then inserted downstream of the 4XT sequence in p-4XT via restriction digest and a two-part ligation, creating the p-8XT plasmid with the 4XT sequence duplicated (Supplementary Table 5). The 12Ig plasmid (p-12XT) was made in a similar fashion. First, the 4XT sequence was PCR-amplified twice, once with primers adding a Kpn2I site to the 5′ end and a SpeI site to the 3′ end; and another time with primers adding a NheI site to the 5′ end and maintaining a stop codon and a BamHI site at the 3′ end (Supplementary Table 6). The resulting PCR amplicons and p-4XT were digested with the corresponding restriction enzymes and ligated in a three-part reaction, yielding the p-12XT plasmid with a triplicated 4XT sequence (Supplementary Table 5).

**Bioproduction in shake flask cultures**. Overnight seed cultures of 50 mL TB medium were inoculated with single colonies carrying the desired construct (Supplementary Table 7). These seed cultures were then used to inoculate cultures of 500 mL TB in 2 L Erlenmeyer flasks at an initial $OD_{600}$ of 0.08. Cultures were placed on reciprocal shakers at 350 rpm at 37 °C until $OD_{600}$ reached 3.0, at which point the corresponding inducer was added (0.2% arabinose for p-1-4XT-1$_B$ and 1 mM IPTG for p-4XT, p-8XT, and p-12XT). Cultures were then continued at 37 °C for 20 h.

**Bioproduction in fed-batch bioreactors**. Both titin monomer and polymer were ultimately produced in 2 L fed-batch bioreactors (Bioflo120, Eppendorf). Transformants containing p-1-4XT-1$_L$ or p-4XT were cultured overnight in 50 mL TB medium at 37 °C on an orbital shaker. The overnight cultures were then used to inoculate an autoclaved 2 L Bioflo120 heat-blanketed bioreactor containing 1.5 L M9 glucose medium with tryptone supplement (see above). Antifoam 204 was added as needed to minimize foaming (approximately 0.01% v/v). Agitation and airflow were regulated to maintain approximately 70% dissolved oxygen (DO). After consumption of the initial 0.5% w/v glucose (as judged by ΔDO), a sterile substrate feed (20% w/v glucose, 48 g/L tryptone, and 10 g/L $MgSO_4 \cdot 7H_2O$) was initiated to maintain a linear growth rate. Reactors were induced at $OD_{600} = 70$ by addition of 1 mM IPTG, and the incubation temperature was reduced to 30 °C. Cultures were collected six hours after induction.

**Protein purification**. The 8 Ig, 12 Ig, and polymer titin were purified by resuspending cell pellets in urea lysis buffer (8 M urea, 300 mM NaCl, 10 mM imidazole, 20 mM $KH_2PO_4$, pH 7.4) at a ratio of 100 mL buffer to 50 g wet cell pellet weight.

The solution was sonicated on ice using a QSonica Q700 sonicator (Qsonica) for 5 min (5 s on, 10 s off). Sonicated lysate was then pelleted by centrifugation at 25,000 × g for 30 min. Cleared supernatant was sonicated for an additional 5 min and then filtered through a 0.45 μm PES filter and applied to a series of six HisTrap HP 5 mL columns on an ÄKTA Pure Chromatography System (GE Healthcare Life Sciences) at a flow rate of 2 mL/min. Loaded columns were washed with two column volumes of lysis buffer, then washed by two column volumes of lysis buffer with 50 mM imidazole, and finally eluted by lysis buffer with 300 mM total imidazole. The ÄKTA chromatography system was controlled by and chromatogram data was acquired using the accompanying UNICORN software (Cytiva).

The titin monomer was purified by dissolving cell pellets in an aqueous lysis buffer (50 mM Tris, 50 mM NaCl, 1 mM PMSF, and 300 μg/mL lysozyme). After stirring for 30 min at 4 °C, 5 mM $MgCl_2$ and 5 μg/mL DNaseI were added, and the mixture was sonicated with stirring on ice for 10 min (5 s on, 10 s off). After sonication, NaCl and imidazole were added to final concentrations of 300 mM and 10 mM, respectively. The mixture was centrifuged at 25,000 × g for 30 min at 4 °C, followed by 75,000 × g for 30 min at 4 °C. Cleared supernatant was then filtered and applied to a series of six HisTrap HP 5 mL columns at 2 mL/min. Loaded columns were washed with 2 column volumes of wash buffer (50 mM Tris, 300 mM NaCl, 10 mM imidazole), then washed with 2 column volumes wash buffer with 50 mM imidazole, and finally eluted with wash buffer with 300 mM imidazole.

After purification by affinity chromatography, the proteins were fully dialyzed to 5 mM ammonium bicarbonate at 4 °C using 10 kDa MWCO snakeskin dialysis tubing (Thermo Fisher Scientific).

**SDS-PAGE**. All SDS-PAGE gels were 1 mm thick, discontinuous with 3% stacking gel, and hand cast at the indicated percentages. Samples were prepared at 1 mg/mL total protein in Laemmli sample buffer (2% SDS, 10% glycerol, 60 mM Tris pH 6.8, 0.01% bromophenol blue, and 100 μM DTT). Gels were run on Mini-PROTEAN Tetra Cells (Bio-Rad) in 1× Tris-glycine SDS buffer (25 mM Tris base, 250 mM glycine, and 0.1% w/v SDS), until just before the dye front exited the gel. For MW estimation, we employed Precision Plus Dual Color Prestained Standards (Bio-Rad) and HiMark Pre-stained Standards (Thermo Fisher). Gels were stained in Coomassie Blue solution (50% v/v methanol, 10% v/v acetic acid, and 1 g/L Coomassie Brilliant Blue) for a minimum of one hour at room temperature with gentle agitation and destained in Coomassie Blue destain buffer (40% v/v methanol and 10% v/v acetic acid) for a minimum of one hour. Gels were imaged using the cSeries Capture Software on an Azure c600 Imager (Azure Biosystems). An unprocessed and uncropped image of the gel in Supplementary Fig. 2a can be found in the Source Data file.

**Analytical SEC**. Protein was concentrated to approximately 10 mg/mL based on absorbance at 280 nm. A Superose 6 Increase 10/300 column (GE Healthcare) was equilibrated with elution buffer (10 mM potassium phosphate, 150 mM NaCl, and pH 7.4), after which 100 μL of the sample were injected onto the column at 0.5 μL/min. The column was then eluted with 1 column volume of elution buffer, and the absorbance of the eluent was measured at 280 nm. Following the same procedure, 100 μL of protein standard mix (MilliporeSigma) and blue dextran (2000 kDa, MilliporeSigma) were separately passed through the column. A calibration curve was prepared by plotting the known MW of the standards against their retention volume ($V_r$) divided by the void volume ($V_0$, blue dextran retention volume). An exponential curve was fit to the calibration data and used to calculate the MW of the titin polymer and monomer based on their measured retention volumes. Polymer number-average MW ($M_n$) was calculated as (1) $M_n = \frac{\sum_i M_i N_i}{\sum_i N_i}$, where ($M_i$) was taken as the calculated MW at a given data point on the polymer chromatogram (including only data from 1 kDa to 5 MDa) and $N_i$ was taken as the measured absorbance at the corresponding data point. Weight-average MW ($M_w$) was calculated as (2) $M_w = \frac{\sum_i N_i M_i^2}{\sum_i N_i M_i}$.

**Circular dichroism**. CD spectra were acquired using the JASCO Spectrum Measurement software on a JASCO J-810 CD spectrometer equipped with a Lauda RM 6 refrigerated circulator and a JASCO PTC-423S peltier. Samples were diluted in 5 mM $NaHCO_3$ and loaded into a 1 mm quartz cuvette (Hellma, Germany). The CD spectra were obtained at 20 °C, scanning in 1 nm steps from 190–260 nm with a 1 nm bandwidth, scanning speed of 100 nm/min, and 2 s response time. Multiple spectra were acquired, each the average of triplicate scans.

**Scanning transmission electron microscopy**. Samples (10 μL) were pipetted onto pure carbon, 400-mesh copper grids (Ted Pella, Inc.) that had been ozone-treated for 15 min using a Novascan PSD Series UV Ozone System (Novascan). After incubating for 5 min, grids were washed with 3 drops of ultrapure water and a 10 μl drop of 0.75% uranyl formate[64]. Grids were then stained by adding a 10 μl drop of 0.75% uranyl formate and incubating for 3 min. The filter paper was used to wick liquids away between each step, and the grids were allowed to air-dry before imaging. Samples were imaged on a scanning transmission electron microscope (STEM, JEM-2100F, JEOL, Japan) set at 200 kV using the GATAN DigitalMicrograph software. STEM images were simultaneously recorded from both a bright-

field (BF) and a high-angle annular dark-field (HAADF) detector. Using ImageJ software (v. 1.52a), fibril cross-sectional diameters were measured approximately every 10 nm along the fibril axis, avoiding regions with ambiguous stain boundaries. A total of 376 diameter measurements were made.

**Fiber spinning.** Fiber spinning was performed by first dissolving lyophilized titin powder in hexafluorisopropanol (HFIP) to 20% w/v. This protein dope was loaded to a 100 μL Hamilton gastight syringe (Hamilton Robotics) fitted with a 23s gauge (116 μm inner diameter and 4.34 cm length) needle. The syringe was fitted to a Harvard Apparatus Pump 11 Elite syringe pump (Harvard Apparatus), and the dope was extruded into a water bath at 5 μL/min. Short segments (~5–10 cm) were then cut from these extruded fibers and carefully extended by hand in water at approximately 1 cm/s to 5× their original length. Extended fibers were removed from the bath and held under tension until visibly dry.

**Light microscopy.** Fiber diameters were measured using images acquired with a Zeiss Axio Observer ZI Inverted Microscope equipped with a phase-contrast 20× objective lens and the Axiovision LE software (Zeiss).

**Scanning electron microscopy.** Following tensile tests, titin fibers were mounted onto a sample holder using double-sided conductive tape (Electron Microscopy Sciences). The sample holder was sputter-coated with a 10 nm gold layer using a Leica EM ACE600 high vacuum sputter coater (Leica Microsystems). Fibers were imaged using a Nova NanoSEM 230 Field Emission Scanning Electron Microscope (Field Electron and Ion Company, FEI) at an accelerating voltage of 7–10 kV using the xT Microscope Control software (FEI).

**Fourier transform infrared spectroscopy.** For secondary structure determination, FT-IR spectra were acquired with a Thermo Nicolet 470 FT-IR spectrometer (Thermo Fisher Scientific) fitted with a Smart Performer ATR accessory with a Ge crystal. Spectra were acquired from 1415–1780 cm$^{-1}$ at 2 cm$^{-1}$ resolution. A total of 254 scans were accumulated for each sample. All recorded spectra were analyzed using Fityk 0.9.8[65]. Baselines were subtracted from all spectra using the built-in Fityk convex hull algorithm. The amide I band (1600–1700 cm$^{-1}$) was deconvolved into a set of eleven Lorentzian peaks centered at 1610, 1618.5, 1624.5, 1632.5, 1642, 1651, 1659, 1666.5, 1678, 1690.5, and 1700 cm$^{-1}$, corresponding to amide I shift characteristic of β-sheet, random coil, α-helix, or β-turn structures[66–68]. Peak areas were integrated, and component percentages were calculated as the component peak area over the sum of all peak areas. Percentages were averaged from measurements of three fibers for each condition (as-spun and post-spin drawn). To directly compare spectra, each individual spectrum was normalized to the highest measured absorbance. Normalized spectra were averaged (three spectra for each condition) and overlaid.

**Polarized Raman spectromicroscopy.** The method reported here is adapted from several previous studies of molecular alignment in spider silk fibers[37,69,70]. Titin fibers were carefully fixed to glass microscope slides with microscale markings to ensure that spectra were acquired at the same location before and after stage rotation. Raman spectra were acquired with a Renishaw RM1000 InVia Confocal Raman Spectrometer (Renishaw) coupled to a Leica DM LM microscope with a rotating stage (Leica Microsystems). Fibers were initially oriented along the x-axis (parallel to the laser polarization). Fibers were irradiated at a fixed point with the 514 nm line of an argon laser with polarization fixed along the x-axis and focused through a 50× objective (NA = 0.75). Spectra were recorded from 1100–1800 cm$^{-1}$ with 1800 lines/mm grating. For each acquisition, a total of 10 spectra were accumulated, each for 10 s. The stage was then rotated to orient fibers along the y-axis with the same laser polarization, and spectra were acquired a second time at the same fixed point. No signs of thermal degradation were apparent, either visually or within recorded spectra. All recorded spectra were analyzed using Fityk 0.9.8[65]. Baselines were subtracted from all spectra using the built-in Fityk automatic convex hull algorithm. For intensity ratio calculations, all spectra were normalized to the intensity of the 1450 cm$^{-1}$ peak, which arises from CH$_2$ bending and is insensitive to protein conformation[69]. For each fiber, the normalized intensity of the peak at 1670 cm$^{-1}$ when oriented along the Y-axis was divided by the normalized intensity of the peak when oriented along the X-axis to give the intensity ratio (3) $I = \frac{Y}{X}$. This procedure was performed on a total of three separate fibers for each condition, and calculated intensity ratios were averaged. Spectra were also averaged and are presented with standard deviations for each point along with the spectra.

**Wide-angle X-ray diffraction data collection.** Synchrotron-based wide-angle X-ray diffraction (WAXD) analysis was performed on the BioCars 14-BM-C beamline at the Advanced Photon Source at Argonne National Laboratory, Argonne, IL. The wavelength of the X-ray beam was 0.886 Å, with fixed energy of 14 keV, and the beam size was 130 × 340 μm$^2$ (horizontal × vertical). 2D diffraction images were recorded using a Pilatus3 S 2 M Detector and the samples-to-detector distance was 200 mm. CeO$_2$ powder was used for the instrument calibration. For X-ray fiber diffraction measurements, the air background was measured first with no sample mounted on the sample stage. Then, bundles of 25 fibers, 1 mm in length and approximately 10 μm in diameter,

were mounted across the opening of a rectangular paper frame. The assembly was loaded onto the sample stage with the fiber axis perpendicular to the X-ray beam, and the exposure time was 60 s to obtain a 2D diffraction image. The obtained diffraction intensities were subtracted by the air background intensity. Multiple images (≥3) were taken to improve the signal/noise ratio.

**Wide-angle X-ray diffraction data analysis.** To analyze the WAXD results, radial and azimuthal 1D profiles were sequentially obtained from the deconvolution of 2D diffraction images using the FIT2D software[71]. The deconvolution and fitting of 1D profiles were performed with the peak analyzer tool in the OriginPro 2016 software (OriginLab, Northampton, MA). The data were fitted with Gaussian functions using nonlinear least-squares fitting. We obtained 1D radial profiles of intensity versus scattering vector $q$ (Å$^{-1}$, radius within the 2D diffraction image) by integrating azimuthally over a sector typically 20–30 degrees wide along either the equator or meridian.

The WAXD analysis assumed an orthorhombic unit cell commonly applied to β-sheet crystallites in semi-crystalline fibers[72–74]. In particular, the 1D radial intensity profile along the equator includes two main equatorial Bragg reflections. Here the innermost equatorial peak is indexed as (200), corresponding to inter-sheet d-spacing along the unit cell a-axis and the outermost equatorial peak is indexed as (120), corresponding to inter-chain d-spacing along the unit cell b-axis (Supplementary Fig. 8)[72–74]. After deconvolution of the 1D profile, we obtained the peak center (PC), full width at half maximum (FWHM), and relative intensity ($I$) of crystalline peaks vs. amorphous components (Fig. 2h, i, Supplementary Table 2). The degree of crystallinity was estimated by dividing the intensities of crystalline peaks by the sum of intensities from crystalline peaks and amorphous components (Supplementary Tables 2, 3) (4) (% Crystallinity = $\frac{I_{\text{Equatorial}(200)} + I_{\text{Equatorial}(120)}}{I_{\text{Equatorial}(200)} + I_{\text{Equatorial}(120)} + I_{\text{EquatorialAmorphous1}} + I_{\text{EquatorialAmorphous2}}}$). The center positions of the (200) and (120) crystalline peaks indicate a-axis inter-sheet d-spacing of 1.08 nm and b-axis inter-chain d-spacing of 0.46 nm, respectively (Supplementary Fig. 8, Supplementary Tables 2, 3). From the center position and FWHM of the (200) and (120) peaks, the Scherrer equation was used to determine the average crystallite size of 1.08 nm along the inter-sheet a-axis and 2.91 nm along the inter-chain b-axis, respectively (Supplementary Fig. 8, Supplementary Tables 2, 3). The Scherrer equation is expressed by (5) $D = \frac{K\lambda}{\beta \cos\theta}$[75] where $D$ is the mean size of the crystallite domains, $K$ is a dimensionless shape factor (with a typical value of 0.9), $\lambda$ is the X-ray wavelength (0.886 nm), $\beta$ is the FWHM value in radians (conversion of the FWHM in our study to radians uses (6) $\beta = 2\arcsin(\frac{\lambda \times \text{FWHM}}{4\pi})$, $\lambda$ is X-ray wavelength), and $\theta$ (°) is the Bragg angle). Conversion of the PC in our study to Bragg angle uses (7) $\theta = \arcsin(\frac{\lambda \times \text{PC}}{4\pi}) \times \frac{360}{2\pi}$, where $\lambda$ is the X-ray wavelength. Because the calculated average crystallite size along the inter-sheet a-axis is the same as the d-spacing, we suggest that β-crystals contain two β-sheets.

To estimate the degree of orientation of the crystallites along the fiber axis, we obtained two azimuthal 1D profiles from the 2D diffraction image (Fig. 2j, k). Figure 2j shows the plot of the diffraction intensity integration as a function of azimuthal angle within the radial range of the equatorial (120) peak, and Fig. 2k is the corresponding plot within the radial range of the equatorial (200) peak. After deconvolution of the 1D profiles, we applied the calculated FWHMs of the crystalline peaks and amorphous components to Herman's orientation function (8) $f_{\text{crystal}} = (3 < \cos^2\varphi > -1)/2$ (Supplementary Tables 2, 3)[39–41]. Here φ is the angle between the c-axis and the fiber axis and $< \cos^2\varphi >$ is obtained based on the equation (9) $< \cos^2\varphi > = 1 - 0.8 < \sin^2(0.4 \times \text{FWMH}_{(200)}) > - 1.2 < \sin^2(0.4 \times \text{FWMH}_{(120)}) >$. The parameter $f_{\text{crystal}}$ is 0 for no preferred orientation and 1 if all crystallites are perfectly aligned[39–41].

**Fiber mechanical testing and cyclic loading measurements.** Segments of post-drawn fibers (20 mm) were carefully laid exactly vertical across a 5 mm (vertical) × 15 mm (horizontal) opening cut into a 20 mm × 20 mm piece of cardstock and fixed with adhesive tape at both ends of the opening. Diameters of mounted fibers were then measured by light microscopy, averaging measurements at three points along the fiber axis. Mechanical properties were measured by axial pull tests on an MTS Criterion Model 41 universal test frame fitted with a 1 N load cell (MTS Systems Corporation). Cardstock holders were mounted between two opposing spring-loaded grips, and the supporting edges were carefully cut. Pull tests were conducted at a relative humidity of 45% and temperature of 22 °C, with a constant crosshead speed of 10 mm/min. Stress-strain curves were recorded by the MTS TestSuite TW Elite software using a 1 N load cell and a sampling rate of 50 Hz. Fiber breaks were recorded when a 90% drop from peak stress was detected. All mechanical properties were automatically calculated by the MTS TestSuite TW Elite software. Ultimate tensile strength was calculated as the maximum measured load over the initial fiber cross-sectional area (A = πr$^2$), as determined from light microscopy diameter measurements. Modulus was calculated as the slope of a linear least-squares fit to the stress/strain data of the initial elastic region. Toughness was calculated as the area under the total stress/strain curve divided by the initial fiber volume (V = πr$^2$h), as calculated from measured initial fiber diameters and set initial gage length of 5 mm. For each protein, a total of 14 fibers were measured in this manner.

For cyclic loading measurements, fibers were prepared and mounted as described above. Fibers were pulled at a rate of 10 mm/min to either: (1) a range of strains, starting near 0% and incrementally increasing until failure; or (2) a fixed,

near-maximal elongation (12% for monomer and 30% for polymer fibers). They were then returned to 0% elongation, treated with 95% humidity air for 1 min, and pulled again. For each test, damping capacity was calculated as the ratio of the energy between the loading and unloading curves over the total energy under the loading curve. Damping energy was calculated as the energy difference between the loading and unloading curves divided by the initial fiber volume, as calculated from measured initial fiber diameters and set initial gage length of 5 mm.

**Molecular dynamics simulation**. The representative simulation volume of the fiber was constructed within the Large-scale Atomic/Molecular Massively Parallel Simulator (LAMMPS)[76] with periodic boundary conditions to simulate bulk behavior. Visual Molecular Dynamics (VMD)[77] was used for the visualization of our systems. Within the periodic box, we assembled an initial molecular structure based on our structural analyses of the real fiber. In detail, we first extracted the all-atomistic structure of the titin monomer (I67–I70, 4 Ig) from protein structure 3B43 in the Protein Data Bank[18]. The psf file was generated using the VMD "Automatic PSF Builder" toolkit. The four linked Ig domains form an oriented fibril. We aligned the 4Ig fibrils along the model y-axis and arranged 2 × 2 stacks of fibrils, with each 4 Ig fibril positioned antiparallel to its flanking 4 Ig fibril and offset by two Ig domains (Supplementary Fig. 12). Then we converted the pdb/psf files into a LAAMPS input file. Sodium ions were then added into the simulation box to neutralize the system. Considering that the titin fibers in our experiments were air-dried before pulling, we did not include explicit water molecules in our model. The CHARMM36 force field was used[78], and the system included 23,312 atoms in total.

After minimizing the system energy for 10,000 time steps in LAMMPS, we first equilibrated the system in the NPT ensemble for 1 ns. Then at room temperature, we compressed the titin fiber in the x and z directions (perpendicular to the fiber axis) from 1 bar to 4000 bars and relaxed it back to 1 bar in 2 ns, followed by equilibration of 5 ns. The temperature and pressure damping parameters were chosen as 0.1 ps and 1 ps, respectively. The applied high pressure is to ensure that side surfaces of the fibrils have close contact with each other so that adjacent Ig-like domains can be paired to pack into the same crystalline domain as indicated by our WAXD results (Supplementary Fig. 8). The β-sheet content in the final model structure was measured to be 22%, in agreement with our FT-IR analysis. Further discussion of the model setup can be found in Supplementary Notes 3 and 4.

To carry out uniaxial tensile tests on the titin fiber, we conducted a non-equilibrium MD (NEMD) simulation in $NP_xP_zT$ ensemble, with x and z directions at atmospheric pressure. Uniaxial strains were applied to the y-direction (fiber axis) with a constant strain rate of $1 \times 10^8$/s using the 'fix deform' command in LAMMPS and a time step of 1 fs. Simulations were carried out for 9 ns, with a final tensile strain of 90%. LAMMPS output the engineering strain and the stress, which consists of a kinetic energy term and the virial term (from interactions between atoms, such as pair, bond, angle, and dihedral contributions). All simulations were run three times.

To measure the intra- and inter-fibril bonded and non-bonded interaction energies, we input the static structures calculated from LAMMPS into the NAMD software[79]. Every structure was equilibrated for 100 ps and run for another 100 ps for energy calculation. We used the 'NAMD Energy' toolkit in VMD to output every intra- and an inter-fibril interaction term. The hydrogen bonds were calculated using the 'Hydrogen Bonds' toolkit in VMD, with only polar atoms (N, O, S) considered. The donor-acceptor distance and angle cutoffs were 3.8 Å and 30 deg, respectively. The salt bridges were calculated using the 'Salt Bridges' toolkit in VMD, with an O-N cutoff distance of 3.5 Å. The atomic stress was output from the 'stress/atom' command in LAMMPS, which also consists of the kinetic energy term and the virial term. During the simulation and measurement, the Lennard Jones cutoff was set as 12.0 Å, and the Particle Mesh Ewald method was used to account for electrostatic energy. Throughout all the simulations, the time step was 1 fs.

**Reporting summary**. Further information on research design is available in the Nature Research Reporting Summary linked to this article.

## Data availability

The 4Ig titin and SI structures were acquired using codes 3B43 [https://doi.org/10.2210/pdb3B43/pdb] and 6QAZ [https://doi.org/10.2210/pdb6QAZ/pdb], respectively, on the Protein Data Bank (https://www.rcsb.org). The data underlying Figs. 2a, e–k, 3a–g, 4a, e, f and Supplementary Figs. 1, 2, 4, 6, 7, 9–11, and 13 are provided in the Source Data file. Other data generated in this study are provided within the paper or the Supplementary Information file. Source data are provided with this paper.

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

## Acknowledgements

We thank Michael Lee for assistance with making buffers and SDS-PAGE gels. We thank Srikanth Singamaneni and Rohit Gupta for assistance with Raman spectromicroscopy. We thank Drs. Irina Kosheleva, Vukica Srajer, and Robert Henning at the BioCARS 14 BM-C Beamline of Argonne National Laboratory for their assistance in setting up the WAXD experiments. We thank Thomas Irving and Weikang Ma at the BioCAT beamline of Argonne National Laboratory for their suggestions regarding WAXD experiments. This work was supported by the Office of Naval Research under the award number (N000141912126) and an Early Career Faculty grant from NASA's Space Technology Research Grants Program (NNX15AU45G to FZ). Use of the Advanced Photon Source was supported by the U.S. Department of Energy, Basic Energy Sciences, Office of Science, under Contract No. DE-AC02-06CH11357. The use of BioCARS was also supported by the National Institutes of Health, National Institute of General Medical Sciences grant 1R24GM111072.

## Author contributions

F.Z. and C.H.B. conceived the project. C.H.B. performed plasmid construction, strain engineering, cell culture, protein expression, protein purification, SDS-PAGE, analytical SEC, fiber spinning, light microscopy, FT-IR, polarized Raman spectromicroscopy, fiber mechanical testing, and associated analysis. C.J.S. performed plasmid construction, cell culture, protein expression, protein purification, SDS-PAGE, CD, STEM imaging, fiber spinning, light microscopy, SEM imaging, fiber mechanical testing, and associated analysis. A.W. and S.K. performed MD simulations and associated analysis. Y.Z. and Y.S.J. performed WAXD and associated analysis. X.C. and X.M. performed cell culture, protein expression, protein purification, and SDS-PAGE. J.L. performed fiber spinning, light microscopy, and fiber mechanical testing. J.G. provided advice and data analysis. C.H.B., C.J.S. and F.Z. prepared the manuscript with comments from all authors.

## Competing interests

C.H.B., C.J.S. and F.Z. have filed a provisional patent application (# 63/113267) based on this work. All other authors declare no competing interests.
