## [Peer Review File · Nature Communications]

REVIEWER COMMENTS

Reviewer #1 (Remarks to the Author):

The authors have developed a process for microbial production of muscle titin-like polymers and demonstrated through mechanical testing that they have exceptional properties. The combination of high modulus and high extensibility result in a remarkable toughness that exceeds that of natural titin and most other known materials. The authors also demonstrate that the polymer shares some of the features of muscle titin, such as strain-dependent damping. A mechanistic model for the mechanical behavior of the polymer is presented.

I found the presentation of the results clear and accessible. My only suggestion would be to include information on the length of the fibers produced and tested. An addition of clamp-to-clamp length on supplementary table 1 would be useful.

Reviewer #2 (Remarks to the Author):

The paper by Bowen et al describes a new approach to produce titin-derived ultra-high molecular weight (UHMW) materials with high mechanical resistance. The work is complemented by molecular simulations, that seem to highlight the importance of inter-chain interactions in defining the mechanical properties of the fibers.

While the findings are potentially very interesting, I have strong concerns in particular on the computational approach:

- The procedure to build the assembly needs to be justified. It seems it was done by just compressing the chains in the initial model along two directions and in a very short simulation time: why would this lead to a meaningful assembly? Normally protein assemblies are built using methods such as homology modelling (if similar templates are available) and/or protein-protein docking and/or integrative modelling approaches, with all these driven by a mixture of physics-based principles and constraints derived from experimental techniques. The approach used in the present paper needs to be justified and its reliability compared to the other techniques. The resulting protein-protein interfaces need also to be analysed and described: what types of interactions are involved? How extensive the interface area?
- The mechanical properties of Ig-like domains (alone or in assemblies, see e.g. 10.1074/jbc.M112.355883) have been vastly characterised in the literature, both with experimental (in particular single-molecule force spectroscopy) and computational techniques (in particular steered MD), but none of these works seem to be really discussed here. The Authors need to discuss their findings in light of previous research, especially from the Fernandez and Schulten groups.
- The computational method used for the actual pulling of the system needs to be described more. What was the strain applied to? What was its magnitude? How does this approach compare to constant velocity Steered MD? For how long the pulling simulations were run?
- Many important simulation parameters are missing: was explicit solvent used and if yes which model? How big was the simulation box and how was the optimal size determined? How many atoms in total? How were the non-bonded interactions calculated and what cutoff was used? What the size of the time step?

Reviewer #3 (Remarks to the Author):

The MW of Titin is > 3 M Da and is monodisperse, whereas the products here are highly polydisperse ranging from 300 K to ~ 2 MDa. The paper would be much stronger if they could purify fractions of different MWs and then make fibers and relate their mechanical properties to MW.

How much of the effects seen are due to processing parameters versus molecular properties? Is unclear, which would require systematic study of the effect of MW on material properties.

“Meanwhile, fibers spun in an identical manner from the low MW 4I exhibited dramatically lower strength (-60%), modulus (-38%), breaking strain (-57%), and toughness (-85%), when compared to the UHMW fibers” — these differences look more impressive when expressed as a percentage but are on an absolute scale not that impressive, especially given the large difference in MW.

Figure 3a, there are 14 different high MW polymer fibers with a range of stress-strain curves. How are these samples different? Why do they have such different properties? The worst of these and the best of the monomers are not that different?. The wide variance in their properties needs explanation and is of concern.

Figure 3d-g: there are no data for monomer control?

“Furthermore, these toughness measures even exceed those of many of the toughest synthetic and natural materials, and far exceed those of traditional microbial materials- which materials are they referring to?”

Overall this paper uses an innovative technology developed by this group, but this has been demonstrated previously—albeit with another system, so that this is no longer new. The poor control of MW distribution of the polymers, and incomplete characterization of the MW distribution —with an uncharacterized UHMW fraction— weakens the paper, and the smaller than expected effect of MW on some of the mechanical properties make the conclusions somewhat murky.

Reviewer #1 (Remarks to the Author):

The authors have developed a process for microbial production of muscle titin-like polymers and demonstrated through mechanical testing that they have exceptional properties. The combination of high modulus and high extensibility result in a remarkable toughness that exceeds that of natural titin and most other known materials. The authors also demonstrate that the polymer shares some of the features of muscle titin, such as strain-dependent damping. A mechanistic model for the mechanical behavior of the polymer is presented.

I found the presentation of the results clear and accessible. My only suggestion would be to include information on the length of the fibers produced and tested. An addition of clamp-to-clamp length on supplementary table 1 would be useful.

We thank the reviewer for the compliments on the work, as well as the suggestion to provide further details about the length of the fibers produced and tested. Each extrusion used roughly 10-15 μ l of high concentration protein dope and produced pre-drawn fibers that were several meters in length. Based on these spinning yields and the observed protein production yields of our engineered bacteria, we estimate that 1 L of shake flask batch culture could produce approximately 250 meters of fiber in a continuous spinning process. The clamp-to-clamp gauge length used for testing all of the fibers in this study was 5 mm; we have added that information to the caption for Supplementary Table 1 as requested.

Reviewer #2 (Remarks to the Author):

The paper by Bowen et al describes a new approach to produce titin-derived ultra-high molecular weight (UHMW) materials with high mechanical resistance. The work is complemented by molecular simulations, that seem to highlight the importance of inter-chain interactions in defining the mechanical properties of the fibers.

While the findings are potentially very interesting, I have strong concerns in particular on the computational approach:

(a) The procedure to build the assembly needs to be justified. It seems it was done by just compressing the chains in the initial model along two directions and in a very short simulation time: why would this lead to a meaningful assembly? Normally protein assemblies are built using methods such as homology modelling (if similar templates are available) and/or protein-protein docking and/or integrative modelling approaches, with all these driven by a mixture of physics-based principles and constraints derived from experimental techniques. The approach used in the present paper needs to be justified and its reliability compared to the other techniques. The resulting protein-protein interfaces need also to be analyzed and described: what types of interactions are involved? How extensive the interface area?

We appreciate the reviewer's concerns and offer the following further explanation. Regarding our model, we first assumed that the titin domains do not unfold appreciably during the fiber spinning process, as proteins are known to remain folded even under high shear flow^{1,2}. Homology modelling is a usual way to predict the 3D structures of a single protein from its sequence, based on known protein structural templates. In Protein Data Bank, we can find a heterodimer structure of two Ig units³ (PDB code 2WWM). The X-ray crystal structure verifies that the complex is held together by a mixture of hydrogen bonds, salt bridges, and hydrophobic interactions³. However, we cannot find suitable templates of a quaternary complex of titin chains.

Next, we did in fact use the protein docking GRAMM-X web server⁴ to establish the initial assembly of two titin chains (Fig. R1a). However, the energy minimized structures do not result in conformations with aligned chains as seen in the experimental data of our fibers. This is likely because the docking results obtain a local minimum by minimizing non-bonded energies, electrostatic, VDW, and solvation contributions⁵ and cannot take into account the surrounding protein chains and mechanical microenvironment during spinning, which can induce packing and orientation. Specifically, high shear flow can significantly impact the supramolecular assembly of proteins. For example, amyloid proteins can transfer from spherical aggregates in low shear flow ($\dot{\gamma} \sim 40/s$) to thick fibers in high shear flow ($\dot{\gamma} \sim 400/s$)⁶. The shear flow in our experiment is calculated as $\dot{\gamma} \sim 500/s$, therefore we assume that the titin chains are likely to be well-aligned along the fiber axis. This assumption is supported by our experimental results from Raman and X-ray diffraction analysis. This, along with the observed large initial modulus of our real titin fibers, leads us to believe that the bending motion of individual titin chains is likely hindered due to lateral packing into bundles through close interactions between chain surfaces. Hence, we applied a lateral pressure to compress the initially loosely assembled titin fiber in our simulation. Additionally, we also noticed the importance of the alignment of single titin chains in the simulated titin fiber. The staggered assembly pattern allows the shear force transmission between adjacent chains, which ensures the high stiffness and toughness of the bulk fiber. In comparison, if the chains stack face-to-face or in any other orientation, then the stress cannot transmit across the fiber, resulting in a very weak fiber (figure (c) below). Therefore, we chose a representative model with two Ig domains stacked into a well-packed configuration. The inter-fibril interaction energy calculated from this configuration (-1415 kcal/mol, Fig. R1b) is more favorable than the structures obtained from GRAMM-X (-1009 kcal/mol, Fig. R1a).

Fig. R1. (a) Equilibrated conformation calculated from GRAMM-X web server. (b) Aligned conformation from our model. (c) Tensile stress-strain curve of face-to-face stacked titin fiber.

In reality, the overlap geometry may be different from our simulation, especially considering the many factors in the fiber fabrication process. However, our model is our current best attempt at heuristically capturing the key physics of the actual spinning process and incorporating our current knowledge of the titin fiber's molecular scale structure. Certainly, we hope that more details will be added and further studied in the future. Additionally, a similar approach based on alignment was adopted by co-author Keten in prior work that focused on building the first atomistic model of spider silk, which has been widely adopted in the field⁷.

Regarding the reviewer's concern on interactions between protein-protein interfaces, we can provide the following further explanation. Between interfaces of equilibrated titin chains, there exist Vdw and electrostatic interactions, which change slightly during the fiber stretching. We also calculated the solvent accessible surface (SASA) of a titin chain buried in surrounding titin chains using VMD⁸. SASA of the single chain is S_1 , SASA of

the single chain with surrounding chains is S_2 , and SASA of the complex without the single chain is S_3 . The single chain interaction area ratio is defined by $(S_3+S_1-S_2)/2S_1^9$, which is calculated as 74% for our titin. In contrast, if the chains are not compressed during equilibration, the interaction area ratio is calculated as 52%. Hence, the titin chains in our model interact extensively with other chains.

We have added the above discussion as Supplementary Note 3.

(b) The mechanical properties of Ig-like domains (alone or in assemblies, see e.g. 10.1074/jbc.M112.355883) have been vastly characterised in the literature, both with experimental (in particular single-molecule force spectroscopy) and computational techniques (in particular steered MD), but none of these works seem to be really discussed here. The Authors need to discuss their findings in light of previous research, especially from the Fernandez and Schulten groups.

We thank the reviewer for reminding us about the works from the Fernandez and Schulten groups. We have added some of the relevant citations in the revised draft. The stretching behavior of a single titin chain with multiple domains has been widely studied, both with AFM experiments¹⁰⁻¹⁵ and steered MD (SMD) simulations^{16,17}. Under tensile force, the force-extension curve of each titin domain follows a worm-like chain form, with an initial slow increase of force followed by a steep increase of force. The unfolding of multiple domains on a titin chain results in a characteristic saw-tooth pattern. For a titin chain with multiple Ig domains, from simulation¹⁷, it is shown that the initial slow force increase is largely because of the bending degree of freedom between adjacent domains. After titin domains are aligned, the force quickly increases. When the first titin domain begins to unfold, the force drops significantly. While the whole domain unfolds, the chain extends following the worm-like chain curve and the force increases to a peak, which is followed by another sudden drop due to the unfolding of a second domain¹⁵. A parallel dimeric titin chain was shown to unfold with a similar force-extension pattern but with a larger peak force when studied using AFM¹⁰.

In contrast, for the biosynthetic, macroscale titin fiber described in this work, the bulk diameter is ~ 10 μm , meaning that we cannot consider it only as a single titin chain or even as a parallel dimeric titin chain. Therefore, our model features tightly packed titin chains and a simulation box that employs periodic boundary conditions, which can give us the bulk properties of the macroscale titin fiber. From our experimental and simulation data, the initial slope of the stress curve is very steep, which likely originates from the fact that the bending degree of freedom of the single titin chains is restrained due to interactions with other chains. More importantly, in the bundled titin fiber, the staggered alignment of single fibers is also pivotal. When we studied models with face-to-face stacking, we observed weak interfaces between chains along the fiber axis. The resulting bundled fiber can never be tough and stiff due to low shear transfer. Therefore, we emphasize that our findings make a step forward in linking the intrinsic properties of Ig

domains to the observed bulk properties of a macro-scale titin fiber.

To address the reviewer's concerns, we have acknowledged the prior work at the beginning of the modeling section of Main Text.

(c) The computational method used for the actual pulling of the system needs to be described more. What was the strain applied to? What was its magnitude? How does this approach compare to constant velocity Steered MD? For how long the pulling simulations were run?

We appreciate the reviewer's concern and have added more details in the MD simulation section of the Methods. Specifically, to conduct the tensile test, we used the 'fix deform' command in LAMMPS, which added a constant uniaxial strain rate ($10^8/s$) to the simulation box in y direction (fiber axis), with a time step of 1 fs. The simulation lasted 9 ns, with a final tensile strain of 90%.

To compare this approach with constant velocity Steered MD (SMD), we first used LAMMPS with the 'fix deform' command to stretch a single titin chain (with the top and bottom of the chain linked across the periodic box boundary, such that this represents an infinite chain with repeats in image cells). The tensile strain rate was $1 \times 10^8/s$, which can be related to a pulling rate of 1.2 m/s that is comparable to previous studies with MD. Fig. R2a is the representative single titin fibril tensile force-displacement curve we calculated using this approach. The overall shape of the curve matches the saw-tooth pattern from experiments and other SMD simulations for single chains^{14,17}, with gradual increases of force and sudden drops after peak force that indicate the cooperative unfolding of titin domains. The distance to fully unfold I67, the first Ig domain to fully unfold, is about 25 nm (Fig. R2a); this distance matches well with previous SMD simulations¹⁷. The sequential unfolding of domains, shown in Fig. R2c, is also similar to previous SMD simulations. Furthermore, to validate our approach, we also used SMD in NAMD to pull a single titin chain with a pulling velocity of 1.2 m/s. Our measured curve of tensile force using SMD (Fig. R2b) matches well the result measured with LAMMPS.

Fig. R2. (a) Tensile force-displacement curve of a single titin fibril simulated by LAMMPS. (b) Tensile force-displacement curve of a single titin fibril simulated by NAMD SMD. (c) Schematic of the stretching of a single titin fiber.

We have added the above discussion as Supplementary Note 4.

(d) Many important simulation parameters are missing: was explicit solvent used and if yes which model? How big was the simulation box and how was the optimal size determined? How many atoms in total? How were the non-bonded interactions calculated and what cutoff was used? What the size of the time step?

We appreciate the reviewer's concern and have added the following details in the Molecular dynamics simulation section of the Methods and Supplementary Fig. 12. Considering that the titin fibers in our experiments were air dried before pulling, we did not include explicit water molecules in our model. In the system, there are 23,312 atoms in total. In Supplementary Fig. 12, we displayed the box size of the starting configuration of the titin fiber and its configuration after equilibration. After equilibration, dimensions of the simulation box are $4.0 \times 13.8 \times 4.6$ nm (x, y, z). We chose this box size to minimize the number of particles simulated (due to limited computational resources) while still yielding a representative system that is informative and reflects the influences of titin interfacial interactions, intrinsic structure, and fibril alignment on the final mechanical properties of the bulk titin fiber. To calculate the non-bonded interactions, we further equilibrated the titin system using NAMD software and measured the interaction terms

using the 'NAMD Energy' toolkit in VMD software. During the simulation and measurement, the Lennard Jones cutoff was set as 12.0 Å, and the Particle Mesh Ewald method was used to account for electrostatic energy. Throughout all the simulations, the time step was 1 fs.

Reviewer #3 (Remarks to the Author):

The MW of Titin is > 3 M Da and is monodisperse, whereas the products here are highly polydisperse ranging from 300 K to ~2 MDa. The paper would be much stronger if they could purify fractions of different MWs and then make fibers and relate their mechanical properties to MW.

How much of the effects seen are due to processing parameters versus molecular properties is unclear, which would require systematic study of the effect of MW on material properties.

“Meanwhile, fibers spun in an identical manner from the low MW 4Ig exhibited dramatically lower strength (-60%), modulus (-38%), breaking strain (-57%), and toughness (-85%), when compared to the UHMW fibers” — these differences look more impressive when expressed as a percentage but are on an absolute scale not that impressive, especially given the large difference in MW.

We appreciate the reviewer's concern regarding the impact of MW on the titin material properties and the suggestion to demonstrate the correlation between mechanical properties and MW. We felt that the clearest way to show this relationship might be to produce, purify, and process titin proteins with well-defined MWs. Consequently, for this analysis we took a genetic approach to changing the MW of the titin by creating proteins with exactly two and three repeats of the original 4Ig monomer (hereafter termed 8Ig and 12Ig, respectively). We cloned, expressed, purified, and spun these 8Ig and 12Ig proteins using the same methods as before. Fiber mechanical properties were measured in parallel with 4Ig. Results from 4Ig, 8Ig, and 12Ig fibers revealed a positive correlation between MW and mechanical performance that agrees well with other types of fiber materials.¹⁸⁻²³ We have added this data (Supplementary Figure 9) and a discussion of our findings to the manuscript (lines 222-232).

Furthermore, our data demonstrated that when compared to the 12Ig (a relatively high MW (128 kDa) and repetitive protein), the titin polymer synthesized from our in vivo polymerization still offers far greater ultimate strength (79% higher) and toughness (85% higher). Equally important, the in vivo polymerization approach eliminates the risks of genetic instability associated with repetitive UHMW proteins from genetic assembly, further demonstrating the value of the in vivo polymerization platform.

We also appreciate the reviewer's concern regarding the potential effects of processing

conditions on fiber mechanical properties. We have taken every measure to control each pertinent variable as tightly as possible. We have even used blinded analysis and confirmed some of our findings with other equipment and replication by multiple researchers to ensure that our process is truly as consistent as we can make it.

Figure 3a, there are 14 different high MW polymer fibers with a range of stress-strain curves. How are these samples different? Why do they have such different properties? The worst of these and the best of the monomers are not that different? The wide variance in their properties needs explanation and is of concern.

When spinning these fibers, each extrusion typically produces meters of pre-drawn fiber. These 14 UHMW polymer fibers are from different sections of those extruded fibers. Clarification of this detail has been added to the Materials and Methods section of the text. Admittedly, variations between measurements are essentially unavoidable when manual drawing, mounting, and measuring are involved. Still, we assure the reviewer that extreme care was taken to minimize error wherever possible and that the variance in our fiber mechanical properties is consistent with that reported in many works on fiber materials²³⁻²⁹.

Even with such variance, our conclusion that the polymer fibers have superior mechanical properties to the 4Ig monomer fibers is still valid. The mean strength of the polymer fibers is 2.5 times that of the monomers; the median strength of the polymers is 2.6 times that of the monomers; and the strength data for the two fibers were found to be significantly different using an unpaired two-tailed t-test ($p < 0.001$; $Df = 26$). Comparisons of the other mechanical properties yield similar results, with unpaired two-tailed t-tests indicating significant differences in all cases ($p < 0.001$; $Df = 26$). To help readers better recognize the significance of these differences, indicators of significance have been incorporated into the figures. We thank the reviewer for these suggestions.

Figure 3d-g: there are no data for monomer control?

We thank the reviewer for bringing this to our attention. We have performed additional experiments to measure the loading/unloading curves with increasing strain and consecutive loading/unloading cycles with humidity treatment using the 4Ig fibers (Supplementary Fig. 11). The 4Ig fibers display similar damping behavior as the polymer fibers. While they are resilient and demonstrate a low damping capacity at low strains, their damping capacity rapidly increases as the strain increases. When pulled to a near maximal strain (12% for the monomer fibers), the fibers were irreversibly deformed upon relaxation. However, exposure to humid air caused the fibers to contract back to their original lengths, from which they could be pulled again. Repeating this load, unload, and humidity treatment cycle revealed that the monomer fibers, similar to those made from the polymer, could endure several cycles of stress and recovery with negligible changes in

their damping capacity in ensuing pulls. However, the monomer fibers exhibited greatly reduced damping energy, which we attribute to their overall lower toughness due to the smaller MW of their protein constituents. We have added a brief discussion of these results to the main text and displayed the results in Supplementary Fig. 11.

“Furthermore, these toughness measures even exceed those of many of the toughest synthetic and natural materials, and far exceed those of traditional microbial materials—which materials are they referring to?”

The materials referred to are presented in Fig. 3c. Supplementary Table 8 also presents the complete list of the synthetic, natural, and microbial materials to which we compare our UHMW titin fibers.

Overall this paper uses an innovative technology developed by this group, but this has been demonstrated previously—albeit with another system, so that this is no longer new. The poor control of MW distribution of the polymers, and incomplete characterization of the MW distribution—with an uncharacterized UHMW fraction—weakens the paper, and the smaller than expected effect of MW on some of the mechanical properties make the conclusions somewhat murky.

We appreciate the reviewer's concerns and believe that performing the suggested investigations into the correlation between MW and mechanical properties and use of the 4Ig monomer fiber as a control have strengthened our conclusions and provide additional evidence for the value of the microbial synthesis platform. The reviewer is correct that the in vivo protein polymerization strategy yields a polydisperse product; however, we would note that the polydispersity is similar to that of traditional organic polymers³⁰ and that the microbial production of a monodisperse UHMW titin protein through genetic assembly is thus far unreported and likely unattainable due to genetic instability. In regards to novelty beyond development of the platform, we believe that this work significantly pushes the envelope of the in vivo polymerization platform by demonstrating production of an entirely unique macro-scale material based on muscle titin—a protein that has not previously been employed for production of macroscale, engineered materials. This material has a wide range of potential applications, and further optimizations of the polymerization are likely to improve mechanical properties and productivity, making practical applications even more likely. Furthermore, the findings of the structural analyses and modeling suggest unique energy dissipation mechanisms and structure-property relationships that we believe could guide the future design of additional high-performance protein-based materials—all facilitated by the in vivo polymerization platform, the versatility of which has been further demonstrated by this work.

Response References:

1. Jaspe, J. & Hagen, S. J. Do Protein Molecules Unfold in a Simple Shear Flow? *Biophys. J.* **91**, 3415–3424 (2006).

2. Phillips, J. C. Scalable molecular dynamics with NAMD. *J. Comput. Chem.* **26**, (2005).
3. Pernigo, S. *et al.* Structural insight into M-band assembly and mechanics from the titin-obscurin-like-1 complex. *Proc. Natl. Acad. Sci.* **107**, 2908 LP – 2913 (2010).
4. Tovchigrechko, A. & Vakser, I. A. GRAMM-X public web server for protein–protein docking. *Nucleic Acids Res.* **34**, W310–W314 (2006).
5. Rosell, M. *et al.* Integrative modeling of protein-protein interactions with pyDock for the new docking challenges. *Proteins Struct. Funct. Bioinforma.* **88**, 999–1008 (2020).
6. Foderà, V., Pagliara, S., Otto, O., Keyser, U. F. & Donald, A. M. Microfluidics Reveals a Flow-Induced Large-Scale Polymorphism of Protein Aggregates. *J. Phys. Chem. Lett.* **3**, 2803–2807 (2012).
7. Ketten, S. & Buehler, M. J. Nanostructure and molecular mechanics of spider dragline silk protein assemblies. *J. R. Soc. Interface* **7**, 1709–1721 (2010).
8. Knapp, B., Lederer, N., Omasits, U. & Schreiner, W. vmdICE: A plug-in for rapid evaluation of molecular dynamics simulations using VMD. *J. Comput. Chem.* **31**, 2868–2873 (2010).
9. Mishra, S. Computational prediction of protein-protein complexes. *BMC Res. Notes* **5**, 495 (2012).
10. Garcia-Manyes, S., Badilla, C. L., Alegre-Cebollada, J., Javadi, Y. & Fernández, J. M. Spontaneous Dimerization of Titin Protein Z1Z2 Domains Induces Strong Nanomechanical Anchoring. *J. Biol. Chem.* **287**, 20240–20247 (2012).
11. Marszalek, P. E. *et al.* Mechanical unfolding intermediates in titin modules. *Nature* **402**, 100–103 (1999).
12. Oberhauser, A. F., Hansma, P. K., Carrion-Vazquez, M. & Fernandez, J. M. Stepwise unfolding of titin under force-clamp atomic force microscopy. *Proc. Natl. Acad. Sci.* **98**, 468 LP – 472 (2001).
13. Oberhauser, A. F., Marszalek, P. E., Erickson, H. P. & Fernandez, J. M. The molecular elasticity of the extracellular matrix protein tenascin. *Nature* **393**, 181–185 (1998).
14. Rico, F., Gonzalez, L., Casuso, I., Puig-Vidal, M. & Scheuring, S. High-Speed Force Spectroscopy Unfolds Titin at the Velocity of Molecular Dynamics Simulations. *Science (80-.)*. **342**, 741 LP – 743 (2013).
15. Rief, M. *et al.* Reversible Unfolding of Individual Titin Immunoglobulin Domains by AFM. *Science (80-.)*. **276**, 1109–1112 (1997).
16. Gao, M., Wilmanns, M. & Schulten, K. Steered Molecular Dynamics Studies of Titin I1 Domain Unfolding. *Biophys. J.* **83**, 3435–3445 (2002).
17. Hsin, J., Strümpfer, J., Lee, E. H. & Schulten, K. Molecular Origin of the Hierarchical Elasticity of Titin: Simulation, Experiment, and Theory. *Annu. Rev. Biophys.* **40**, 187–203 (2011).
18. Yang, Y. J., Holmberg, A. L. & Olsen, B. D. Artificially Engineered Protein Polymers. *Annu. Rev. Chem. Biomol. Eng.* **8**, 549–575 (2017).
19. Wegst, U. G. K., Bai, H., Saiz, E., Tomsia, A. P. & Ritchie, R. O. Bioinspired

- structural materials. *Nat. Mater.* **14**, 23–36 (2015).
20. Omenetto, F. G. & Kaplan, D. L. New Opportunities for an Ancient Material. *Science (80-.).* **329**, 528 LP – 531 (2010).
 21. Ling, S., Kaplan, D. L. & Buehler, M. J. Nanofibrils in nature and materials engineering. *Nat. Rev. Mater.* **3**, 1–15 (2018).
 22. Nunes, R., Martin, J. & Johnson, J. Influence of Molecular Weight and Molecular Weight Distribution on Mechanical Properties of Polymers. *Polym. Eng. Sci.* **22**, 205–228 (1982).
 23. Xia, X.-X. *et al.* Native-sized recombinant spider silk protein produced in metabolically engineered *Escherichia coli* results in a strong fiber. *Proc. Natl. Acad. Sci. U.S.A.* **107**, 14059–14063 (2010).
 24. Peng, Q. *et al.* Recombinant spider silk from aqueous solutions via a bio-inspired microfluidic chip. *Sci. Rep.* **6**, 36473 (2016).
 25. Andersson, M., Jia, Q., Abella, A., Lee, X. & Landreh, M. Biomimetic spinning of artificial spider silk from an extremely concentrated chimeric minispidroin. *Nat. Chem. Biol.* 1–22 (2017). doi:10.1038/nchembio.2269
 26. Heidebrecht, A. *et al.* Biomimetic Fibers Made of Recombinant Spidroins with the Same Toughness as Natural Spider Silk. *Adv. Mater.* **27**, 2189–2194 (2015).
 27. Lewin, M. *Handbook of Fiber Chemistry*. (CRC Press, 1998).
 28. Wu, Y. *et al.* Biomimetic Supramolecular Fibers Exhibit Water-Induced Supercontraction. *Adv. Mater.* **30**, (2018).
 29. Wegst, U. G. K. & Ashby, M. F. The mechanical efficiency of natural materials. *Philos. Mag.* **84**, 2167–2186 (2004).
 30. Chanda, M. & Roy, S. K. *Plastics Technology Handbook* (4th ed.). *CRC Press* (2006). doi:<https://doi.org/10.1201/9781420006360>

REVIEWERS' COMMENTS

Reviewer #2 (Remarks to the Author):

The Authors have addressed all my comments

Reviewer #3 (Remarks to the Author):

The authors have addressed my concerns. I have no further comments.